# Investigation of the relationship between sensorineural hearing loss and associated comorbidities in patients with chronic kidney disease: A nationwide, population-based cohort study

Kun-Lin Wu[1,2,3], Cheng-Ping Shih[4], Jenq-Shyong Chan[1,2], Chi-Hsiang Chung [5,6], Hung-Che Lin[4], Chang-Huei Tsao[5,7], Fu-Huang Lin[6], Wu-Chien Chien[5,6]*, Po-Jen Hsiao [1,2,8,9]*

1 Division of Nephrology, Department of Internal Medicine, Tri-Service General Hospital, National Defense Medical Center, Taipei, Taiwan, Republic of China, 2 Division of Nephrology, Department of Internal Medicine, Taoyuan Armed Forces General Hospital, Taoyuan City, Taiwan, Republic of China, 3 Department of Biomedical Sciences and Engineering, Institute of Systems Biology and Bioinformatics, National Central University, Taoyuan, Taiwan, Republic of China, 4 Department of Otolaryngology-Head and Neck Surgery, Tri-Service General Hospital, National Defense Medical Center, Taipei, Taiwan, Republic of China, 5 Department of Medical Research, Tri-Service General Hospital, Taipei, Taiwan, Republic of China, 6 School of Public Health, National Defense Medical Center, Taipei, Taiwan, Republic of China, 7 Department of Microbiology & Immunology, National Defense Medical Center, Taipei, Taiwan, Republic of China, 8 Big Data Research Center, Fu-Jen Catholic University, Taipei, Taiwan, Republic of China, 9 Department of Life Sciences, National Central University, Taoyuan City, Taiwan, Republic of China

☯ These authors contributed equally to this work.

* a2005a660820@yahoo.com.tw (PJH); chienwu@ndmctsgh.edu.tw (WCC)

**Data Availability Statement:** All data are available from the National Health Insurance Administration, Ministry of Health and Welfare in Taiwan for researchers who meet the criteria for access to

## Abstract

Hearing impairment was observed in patients with chronic kidney disease (CKD). Our purpose was to investigate the relationship between sensorineural hearing loss (SNHL) and associated comorbidities in the CKD population. We conducted a retrospective, population-based study to examine the risk of developing SNHL in patients with CKD. Population-based data from 2000–2010 from the Longitudinal Health Insurance Database of the Taiwan National Health Insurance Research Database was used in this study. The population sample comprised 185,430 patients who were diagnosed with CKD, and 556,290 without CKD to determine SNHL risk factors. Cox proportional hazard regression analysis demonstrated the CKD group had a significantly increased risk of SNHL compared with the non-CKD group [adjusted hazard ratio (HR), 3.42; 95% confidence interval (CI), 3.01–3.90, $p <$ 0.001]. In the CKD group, the risk of SNHL (adjusted HR, 5.92) was higher among patients undergoing hemodialysis than among those not undergoing hemodialysis (adjusted HR, 1.40). Furthermore, subgroup analysis revealed an increased risk of SNHL in patients with CKD and comorbidities, including heart failure (adjusted HR, 7.48), liver cirrhosis (adjusted HR, 4.12), type 2 diabetes mellitus (adjusted HR, 3.98), hypertension (adjusted HR, 3.67), and chronic obstructive pulmonary disease (adjusted HR, 3.45). CKD is an independent risk of developing SNHL. Additionally, hemodialysis for uremia can increase the risk of SNHL.

confidential data. Due to legal restrictions imposed by the government of Taiwan in relation to the "Personal Information Protection Act", data cannot be made publicly available. The contact information for the National Health Insurance Research Database, Taiwan is nhird@nhri.org.tw.

**Funding:** This study was supported by grants from the Research Fund of the Taoyuan Armed Forces General Hospital (AFTYGH-109-009) and Tri-Service General Hospital (TSGH-B-109010).

**Competing interests:** The authors have declared that no competing interests exist.

Cardiovascular, lung, liver, and metabolic comorbidities in CKD patients may further aggravate the risk of SNHL by inter-organ crosstalk. We should pay attention to SNHL in this high-risk population.

## Introduction

Chronic kidney disease (CKD) is characterized by a progressive reduction in the renal function and can affect several organs. CKD is continually becoming more prevalent and a public health issue of global concern [1]. Problems associated with the auditory system are common in patients with CKD and can negatively affect the quality of life [2].

The genetic condition that links the kidneys to the ears is Alport's syndrome. Reportedly, the nephron shares anatomical, physiological, immunological, and pharmacological similarities with the stria vascularis. Epithelial cells of nephron and stria vascularis are in close contact with their vascular supply [3]. The sodium–potassium pump, carbonic anhydrase, calcium-ATPase, and calcium-binding proteins actively transport fluid and electrolytes in both organs [4–6]. Common antigenicity testing has demonstrated antibody deposition in both nephron and stria vascularis [3, 7]. Various drugs act on both organs such as aminoglycosides associated with both nephrotoxic and ototoxic effects [3, 4].

Recently, a high prevalence of CKD has been reported globally [1]. Hearing loss is highly prevalent in patients with CKD compared with the general population [8]. As renal insufficiency progresses, uremic toxin accumulates, which has adverse pathological effects [4]. Furthermore, hearing impairment and uremia have been shown to be associated with each other [9], and can negatively impact the patient's quality of life by limiting communication and thereby introducing a risk of social isolation and emotional difficulties [8, 10].

CKD patients often suffer associated comorbidities such as hypertension, type 2 diabetes mellitus (DM), heart failure (HF), stroke, chronic obstructive pulmonary disease (COPD), and liver cirrhosis [11]. To identify the relationship of developing SNHL among CKD and comorbidities, a population-based retrospective cohort study was conducted using data from the Taiwan National Health Insurance Research Database (NHIRD).

## Materials and methods

### Data sources

The National Health Insurance Program is a universal health-care system which contracts with 97% of the medical providers and covers medical expenses of more than 99% of the 23 million inhabitants of Taiwan. The Bureau of National Health Insurance randomly reviews the records of 1 in 100 ambulatory care visits and 1 in 20 in-patient claims to verify the accuracy of the diagnoses. The accuracy and validity of the diagnoses have been demonstrated [12, 13]. Therefore, the NHIRD was used as the data source. The study used data from 2000 to 2010 from the Longitudinal NHIRD to analyze the relationship of developing SNHL in patients with CKD. The International Classification of Disease (9th Revision) Clinical Modification (ICD-9-CM) codes for diagnoses and procedures and of genders and dates of birth recorded in the NHIRD were used. The study protocol was approved by the Ethics Committee of Human Studies at the Tri-Service General Hospital, Taiwan (TSGH IRB No. B-109-13).

### Study design and participants

A 10-year retrospective cohort study was designed and 986,713 consecutive patients from January 1, 2000 to December 31, 2010 were enrolled. Patients newly diagnosed with CKD (ICD-

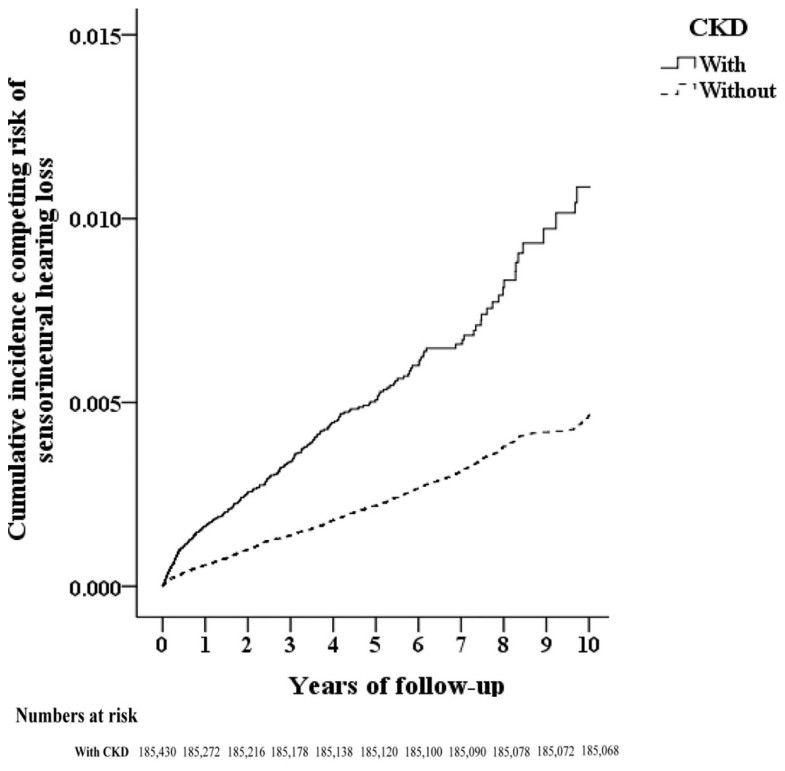

Fig 1. The flowchart of study sample selection.

9-CM codes, 585–586) before 1999, who had received a renal transplantation (ICD-9-CM code, V42.0), who had SNHL (ICD-9-CM code, 389.1) or tinnitus (ICD-9-CM code 388.3) before tracking, who aged <18 years, and who were without tracking or of unknown gender were excluded. After matching the CKD participants with thrice comparison subjects (index year, month, gender and age), 185,430 patients with first diagnosis of CKD and 556,290 participants without CKD were included in the CKD and comparison groups, respectively. The occurrence of SNHL (ICD-9-CM code, 389.1) that was diagnosed by otorhinolaryngologists at least thrice, continued for at least 4 weeks, and tracked until December 31, 2010 (Fig 1). Additionally, a subgroup analysis was conducted for identifying the relationship of developing hearing loss in patients with CKD undergoing and those not undergoing hemodialysis.

The covariates of gender, age groups (18–29, 30–39, 40–49, 50–59, ≥60 years), and insured premium [in New Taiwan Dollars; <18,000, 18,000–34,999, ≥35,000] were analyzed. Baseline comorbidities such as hypertension (ICD-9-CM codes, 401–405), type 2 DM (ICD-9-CM code, 250), HF (ICD-9-CM code, 428), stroke (ICD-9-CM codes, 430–438), chronic obstructive pulmonary disease (COPD) (ICD-9-CM codes, 490–496), and liver cirrhosis (ICD-9-CM code, 571) were included as covariates. Medical histories of aminoglycosides and loop diuretics in the individuals for at least one week were also accessed.

## Statistical analysis

In this population-based, retrospective cohort study, all analyses were performed using the SPSS software version 22 (SPSS Inc., Chicago, Illinois, USA). We present standardized difference and standardized mean difference for categorical and continuous variable distributions,

respectively. Meanwhile, Chi-square ($\chi$2) and t-tests were used. The results are presented as Wald coefficient and hazard ratios (HR) with a 95% confidence interval (CI). Multivariable cox proportional hazards regression analysis was used to determine the risk of developing hearing loss. We also performed a competing-risks regression (Fine-Gray model) because SNHL risk might compete with the risk of death [14]. The cumulative incidence competing risk (CICR) method was used to estimate the difference in the risk of developing hearing loss between the CKD and comparison groups [15]. The significance threshold of p-value was set at 0.05. The 19 weighted indicators of 17 comorbidities were used to calculate the Charlson Comorbidity Index (CCI) [16]. The variables, including CKD, type 2 DM, HF, stroke, COPD, and liver cirrhosis, have been removed to calculate CCI_R.

## Results

In total, 185,430 patients with CKD and 556,290 participants without CKD were enrolled in the study (Fig 1). Compared with the comparison group, the CKD group demonstrated significantly higher rates of hypertension, type 2 DM, HF, and liver cirrhosis ($p < 0.001$) and lower rates of stroke, COPD and Meniere's disease ($p < 0.001$). A higher CCI_R was reported in the CKD group ($p < 0.001$) (Table 1).

After adjusting for variables like age, gender, comorbidities, and drug intake, the highest HR of SNHL was 3.42 times (95% CI = 3.01–3.90) in patients with CKD, followed by that for stroke (HR, 1.52; 95% CI, 1.31–1.77), COPD (HR, 1.21; 95% CI, 1.04–1.53), and liver cirrhosis (HR, 1.19; 95% CI, 0.94–1.55). The HR for aminoglycoside and loop diuretics were lower (Table 2). The cumulative incidence competing risk analysis indicated that patients with CKD had a significantly higher incidence of developing SNHL over time than comparison participants ($p < .001$) (Fig 2).

In the CKD group, 87,361 patients (47.11%) were undergoing hemodialysis and 98,069 (52.89%) were not undergoing hemodialysis. In patients with CKD undergoing hemodialysis, the incidence of SNHL was 133,761.97 per $10^5$ person-years, with these patients having a higher risk of developing SNHL (adjusted HR, 5.92; 95% CI, 3.03–11.79) than those not undergoing hemodialysis (adjusted HR, 1.40, 95% CI = 1.01–3.23) (Table 3).

In the subgroups stratified by gender, age, comorbidities and drug intake, patients with CKD who had comorbid HF, liver cirrhosis, type 2 DM, hypertension, and COPD had higher risks of developing SNHL than those without these comorbidities. In competing risks model, the adjusted HRs of hearing loss were 7.48 and 3.29 in those with and without HF, 4.12 and 3.39 in those with and without liver cirrhosis, 3.98 and 3.24 in those with and without type 2 DM, 3.67 and 3.36 in those with and without hypertension, and 3.45 and 3.35 in those with and without COPD, respectively (Table 4). The increased values of adjusted HRs may indicate the possible influence of CKD and comorbidities for SNHL.

We next focus our investigation on identifying and quantifying the multiplicative interaction of comorbidities on CKD. In Table 5, the highest Wald coefficient and adjusted HR were 49.34 and 7.69 in those with Meniere's disease, 38 and 2.73 in those with stroke, 27.86 and 2.98 in those with liver cirrhosis, 22.57 and 2.6 in those with COPD, 18.16 and 2.37 in those with HTN, 9.39 and 1.92 in those with HF, and 3.8 and 1.31 in those with Type 2 DM, respectively (Table 5). Interestingly, we demonstrated that the interactions of comorbidities on CKD was significant for SNHL.

## Discussion

In this study, a large cohort of patients with newly diagnosed CKD was evaluated for the relationship of developing SNHL. It was observed that the prevalence of CKD was 25% of patients

**Table 1. Characteristics of study in the baseline.**

| Variables | Overall (n = 741,720) | | With CKD (n = 185,430) | | Without CKD (n = 556,290) | | P | Standardized difference | Standardized mean difference |
|---|---|---|---|---|---|---|---|---|---|
| | n | % | n | % | n | % | | | |
| Gender | | | | | | | 0.999 | 0.000 | 0.000 |
| Male | 402,880 | 54.32 | 100,720 | 54.32 | 302,160 | 54.32 | | | |
| Female | 338,840 | 45.68 | 84,710 | 45.68 | 254,130 | 45.68 | | | |
| Age (years) | 67.31 ± 13.55 | | 67.30 ± 14.20 | | 67.31 ± 13.33 | | 0.783 | -1.995 | -0.363 |
| Age groups (yrs) | | | | | | | 0.999 | 0.000 | 0.000 |
| 18–29 | 10,226 | 1.38 | 2,559 | 1.38 | 7,667 | 1.38 | | | |
| 30–39 | 24,428 | 3.29 | 6,107 | 3.29 | 18,321 | 3.29 | | | |
| 40–49 | 63,152 | 8.51 | 15,788 | 8.51 | 47,364 | 8.51 | | | |
| 50–59 | 111,160 | 14.99 | 27,790 | 14.99 | 83,370 | 14.99 | | | |
| ≧60 | 532,744 | 71.83 | 133,186 | 71.83 | 399,558 | 71.83 | | | |
| Insured premium (NT$) | | | | | | | <0.001*** | -0.032 | -0.002 |
| <18,000 | 732,832 | 98.80 | 183,435 | 98.92 | 549,397 | 98.75 | | | |
| 18,000–34,999 | 7,780 | 1.05 | 1,788 | 0.96 | 5,992 | 1.08 | | | |
| ≧35,000 | 1,158 | 0.16 | 207 | 0.11 | 951 | 0.17 | | | |
| Comorbidities | | | | | | | | | |
| HTN | 156,363 | 21.08 | 42,234 | 22.78 | 114,129 | 20.52 | <0.001*** | 0.023 | 0.001 |
| T2DM | 144,996 | 19.55 | 63,178 | 34.07 | 81,818 | 14.71 | <0.001*** | 0.202 | 0.001 |
| HF | 38,776 | 5.23 | 21,441 | 11.56 | 17,335 | 3.12 | <0.001*** | 0.084 | 0.001 |
| Stroke | 69,462 | 9.36 | 13,180 | 7.11 | 56,282 | 10.12 | <0.001*** | -0.030 | -0.001 |
| COPD | 66,119 | 8.91 | 11,927 | 6.43 | 54,192 | 9.74 | <0.001*** | -0.003 | -0.001 |
| Liver cirrhosis | 41,886 | 5.65 | 10,910 | 5.88 | 30,976 | 5.57 | <0.001*** | 0.003 | 0.001 |
| Meniere's disease | 13,890 | 1.87 | 1,434 | 0.77 | 12,456 | 2.24 | <0.001*** | -0.015 | <0.001 |
| CCI_R | 1.24 ± 2.06 | | 2.63 ± 1.61 | | 0.77 ± 1.98 | | <0.001*** | 1.942 | 0.005 |
| Medications | | | | | | | | | |
| Aminoglycoside | 4,585 | 0.62 | 600 | 0.32 | 3,985 | 0.72 | <0.001*** | -0.003 | -0.001 |
| Loop diuretics | 4,733 | 0.64 | 1,634 | 0.88 | 3,099 | 0.56 | <0.001*** | 0.004 | 0.001 |

P: Chi-square / Fisher exact test on category variables and t-test on continue variables.

*P < 0.05, **P < 0.01

***P < 0.001. CKD: chronic kidney disease, HTN: hypertension, T2DM: type 2 diabetes mellitus, HF: heart failure, COPD: chronic obstructive pulmonary disease.

receiving out-patient care, particularly in male ones. Additionally, CKD itself could be a critical role for developing SNHL. Moreover, organ crosstalk between CKD and several comorbidities such as Meniere's disease, stroke, liver cirrhosis, COPD, hypertension, HF, and type 2 DM were found to increase the interaction of developing SNHL in patients with CKD.

CKD patients are more prone to develop SNHL, which results from the delayed conduction between the auditory nerve and pathway [17, 18]. Uremic toxins can cause serial damage in the cochlea [18–20]. The decrease in the adenosine triphosphatase sodium–potassium pump ($Na^+$–$K^+$–ATPase) activity [4] and amplitudes of cochlear potentials [21], and further reduction in velocity conduction in auditory nerve [22] leaded to hearing impairment. Furthermore, cochlear microcirculation plays an important role in cochlear physiology. Non-conventional risk factors such as chronic inflammation, oxidative stress, asymmetric dimethylarginine, sympathetic nerve hyperactivity, prothrombotic state, and hyperhomocysteinemia cause vascular injury and endothelial dysfunction in patients with CKD [23–25]. Although hemodialysis is a renal replacement therapy for uremia, it is a risk factor for developing SNHL [26]. Osmotic

**Table 2. Factors of sensorineural hearing loss by using Cox regression and Fine & Gray's competing risk model.**

| | No competing risk in the model | | | | | Competing risk in the model | | | | |
|---|---|---|---|---|---|---|---|---|---|---|
| | Wald | Adjusted HR | 95% CI | 95% CI | *P* | Wald | Adjusted HR | 95% CI | 95% CI | *P* |
| CKD (*Reference: Without*) | 239.14 | 3.19 | 2.80 | 3.64 | <0.001*** | 303.22 | 3.42 | 3.01 | 3.90 | <0.001*** |
| Gender (*Reference: Female*) | 19.14 | 1.30 | 1.16 | 1.45 | <0.001*** | 19.68 | 1.32 | 1.19 | 1.48 | <0.001*** |
| Age groups (yrs) (*Reference: 18–29*) | | | | | | | | | | |
| 30–39 | 1.54 | 0.52 | 0.19 | 1.40 | 0.192 | 1.71 | 0.53 | 0.20 | 1.43 | 0.205 |
| 40–49 | 0.01 | 0.95 | 0.45 | 2.21 | 0.896 | 0.02 | 0.99 | 0.42 | 2.30 | 0.969 |
| 50–59 | <0.01 | 0.99 | 0.44 | 2.24 | 0.968 | <0.01 | 1.05 | 0.46 | 2.37 | 0.931 |
| ≧60 | 0.02 | 0.12 | 0.51 | 2.53 | 0.790 | 0.07 | 1.28 | 0.57 | 2.86 | 0.568 |
| Comorbidities (*Reference: Without*) | | | | | | | | | | |
| HTN | 2.51 | 0.89 | 0.79 | 1.01 | 0.052 | 3.82 | 0.84 | 0.75 | 0.95 | 0.009** |
| T2DM | 4.25 | 0.86 | 0.76 | 1.01 | 0.051 | 4.19 | 0.85 | 0.75 | 1.00 | 0.046* |
| HF | 21.70 | 0.52 | 0.40 | 0.69 | <0.001*** | 22.57 | 0.55 | 0.42 | 0.73 | <0.001*** |
| Stroke | 21.80 | 1.49 | 1.29 | 1.74 | <0.001*** | 26.22 | 1.52 | 1.31 | 1.77 | <0.001*** |
| COPD | 4.02 | 1.22 | 1.04 | 1.45 | 0.026 * | 3.94 | 1.21 | 1.04 | 1.53 | 0.019 * |
| Liver cirrhosis | 0.01 | 1.13 | 0.89 | 1.45 | 0.338 | 0.83 | 1.19 | 0.94 | 1.55 | 0.192 |
| Meniere's disease | 568.85 | 8.51 | 7.11 | 10.17 | <0.001*** | 546.06 | 7.81 | 6.53 | 9.34 | <0.001*** |
| CCI_R (*Reference: Without*) | 8.77 | 0.95 | 0.93 | 0.98 | 0.006** | 16.38 | 0.97 | 0.94 | 1.00 | 0.027* |
| Medications | | | | | | | | | | |
| Aminoglycoside | 1.02 | 0.37 | 0.12 | 1.14 | 0.267 | 1.86 | 0.37 | 0.12 | 1.18 | 0.143 |
| Loop diuretics | 5.51 | 0.19 | 0.05 | 0.75 | <0.001*** | 11.04 | 0.21 | 0.05 | 0.86 | 0.005* |

Adjusted HR (hazard ratio): Adjusted variables listed in the table; CI = confidence interval

*$P$ < 0.05

**$P$ < 0.01

***$P$ < 0.001

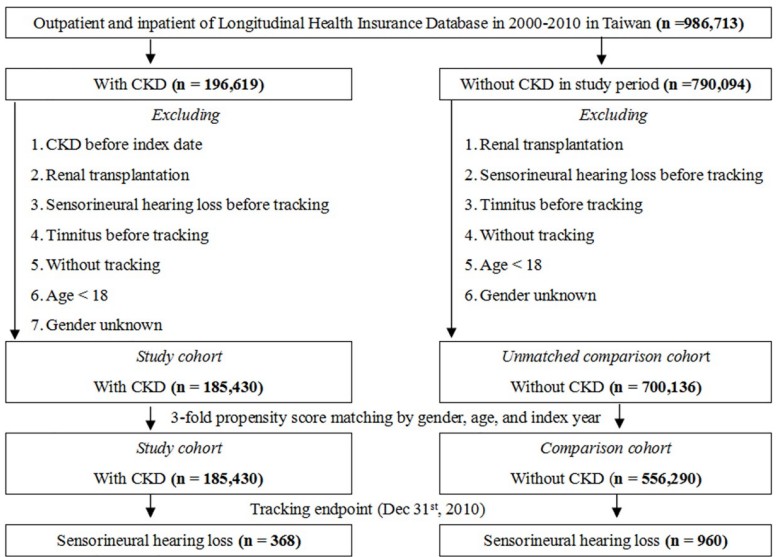

**Fig 2. The cumulative incidence competing risk (CICR) method for the incidence of sensorineural hearing loss among patients aged 18 and over stratified by CKD ($p$ < .001).**

**Table 3. Factors of sensorineural hearing loss stratified by with / without hemodialysis by using Cox regression and Fine & Gray's competing risk model.**

| | Populations | Events | PYs | Rate | No competing risk in the model | | | | | Competing risk in the model | | | | |
|---|---|---|---|---|---|---|---|---|---|---|---|---|---|---|
| | | | | | Wald | Adjusted HR | 95% CI | 95% CI | P | Wald | Adjusted HR | 95% CI | 95% CI | P |
| Without CKD | 556,290 | 960 | 1,168,647.66 | 82.146 | | *Reference* | | | | | *Reference* | | | |
| With CKD | 185,430 | 368 | 334,193.43 | 110.116 | 239.14 | 3.19 | 2.80 | 3.64 | <0.001*** | 303.22 | 3.42 | 3.01 | 3.90 | <0.001*** |
| without hemodialysis | 98,069 | 198 | 200,431.46 | 98.787 | 99.38 | 1.31 | 0.99 | 3.01 | 0.058 | 107.05 | 1.40 | 1.01 | 3.23 | 0.046* |
| with hemodialysis (ESRD) | 87,361 | 170 | 133,761.97 | 127.091 | 485.42 | 5.52 | 2.82 | 10.98 | <0.001*** | 515.62 | 5.92 | 3.03 | 11.79 | <0.001*** |

PYs = Person-years; Rate: per 100,000 PYs; Adjusted HR (hazard ratio): Adjusted variables listed in Table 1; CI = confidence interval

*P < 0.05, **P < 0.01

***P < 0.001; ESRD: end-stage renal disease.

**Table 4. Factors of sensorineural hearing loss stratified by variables listed in the table by using Fine & Gray's competing risk model.**

| Variables | Strarified | With CKD | | | With CKD | | | With CKD *vs.* Without CKD *(Reference)* | | | | |
|---|---|---|---|---|---|---|---|---|---|---|---|---|
| | | Events | PYs | Rate | Events | PYs | Rate | Wald | Adjusted HR | 95% CI | 95% CI | P |
| | Overall | 368 | 334,193.43 | 110.12 | 960 | 1,168,647.66 | 82.15 | 303.22 | 3.42 | 3.01 | 3.90 | <0.001*** |
| Gender | Male | 198 | 174,093.39 | 113.73 | 595 | 629,014.38 | 94.59 | 132.92 | 2.98 | 2.50 | 3.54 | <0.001*** |
| | Female | 170 | 160,100.04 | 106.18 | 365 | 539,633.28 | 67.64 | 177.97 | 4.31 | 3.44 | 5.15 | <0.001*** |
| Age groups (yrs) | 18–29 | 6 | 2,450.49 | 244.85 | 0 | 4,350.17 | 0.00 | 0.011 | ∞ | - | - | 0.897 |
| | 30–39 | 6 | 8,722.15 | 68.79 | 5 | 15,192.58 | 32.91 | 2.266 | 3.37 | 0.77 | 14.82 | 0.724 |
| | 40–49 | 32 | 24,547.77 | 130.36 | 23 | 69,293.14 | 33.19 | 19.729 | 4.30 | 2.34 | 7.89 | <0.001*** |
| | 50–59 | 40 | 54,197.86 | 73.80 | 99 | 305,306.51 | 32.43 | 3.838 | 1.61 | 1.08 | 2.41 | 0.041* |
| | ≧60 | 284 | 244,275.16 | 116.26 | 833 | 744,505.26 | 111.89 | 278.74 | 3.67 | 3.18 | 4.23 | <0.001*** |
| HTN | Without | 216 | 196,821.62 | 109.74 | 677 | 823,506.35 | 82.21 | 184.95 | 3.36 | 2.85 | 3.99 | <0.001*** |
| | With | 152 | 137,371.81 | 110.65 | 283 | 345,141.31 | 82.00 | 121.14 | 3.67 | 2.95 | 4.62 | <0.001*** |
| T2DM | Without | 244 | 226,763.67 | 107.60 | 782 | 902,618.82 | 86.64 | 196.28 | 3.24 | 2.77 | 3.77 | <0.001*** |
| | With | 124 | 107,429.76 | 115.42 | 178 | 266,028.84 | 66.91 | 99.19 | 3.98 | 3.08 | 4.85 | <0.001*** |
| HF | Without | 338 | 302,066.10 | 111.90 | 936 | 1,085,462.92 | 86.23 | 265.05 | 3.29 | 2.87 | 3.76 | <0.001*** |
| | With | 30 | 32,127.33 | 93.38 | 24 | 83,184.74 | 28.85 | 38.87 | 7.48 | 4.08 | 13.78 | <0.001*** |
| Stroke | Without | 329 | 303,790.75 | 108.30 | 793 | 1,031,091.16 | 76.91 | 270.73 | 3.48 | 3.03 | 4.02 | <0.001*** |
| | With | 39 | 30,402.68 | 128.28 | 167 | 137,556.50 | 121.40 | 28.67 | 2.99 | 2.06 | 4.35 | <0.001*** |
| COPD | Without | 342 | 313,272.56 | 109.17 | 830 | 1,024,602.66 | 81.01 | 275.41 | 3.39 | 2.96 | 3.90 | <0.001*** |
| | With | 26 | 20,920.87 | 124.28 | 130 | 144,045.00 | 90.25 | 27.73 | 3.45 | 2.23 | 5.32 | <0.001*** |
| Liver cirrhosis | Without | 344 | 317,482.25 | 108.35 | 915 | 1,108,300.52 | 82.56 | 280.16 | 3.39 | 2.97 | 3.88 | <0.001*** |
| | With | 24 | 16,711.18 | 143.62 | 45 | 60,347.14 | 74.57 | 24.31 | 4.12 | 2.42 | 7.03 | <0.001*** |
| Meniere's disease | Without | 356 | 331,468.23 | 107.40 | 834 | 1,131,989.89 | 73.68 | 308.07 | 3.58 | 3.13 | 4.10 | <0.001*** |
| | With | 12 | 2,725.20 | 440.33 | 126 | 36,657.77 | 343.72 | 1.69 | 1.64 | 1.01 | 3.12 | 0.047* |
| Aminoglycoside | Without | 366 | 332,526.32 | 110.07 | 953 | 1,161,049.27 | 82.08 | 137.94 | 3.41 | 3.00 | 3.92 | <0.001*** |
| | With | 2 | 1,667.11 | 119.97 | 7 | 7,598.39 | 92.12 | 134.65 | 3.33 | 2.83 | 3.83 | <0.001*** |
| Loop diuretics | Without | 367 | 331,401.74 | 110.74 | 957 | 1,157,557.93 | 82.67 | 138.02 | 3.42 | 3.00 | 3.93 | <0.001*** |
| | With | 1 | 2,791.69 | 35.82 | 3 | 11,089.73 | 27.05 | 135.29 | 3.39 | 2.96 | 3.86 | <0.001*** |

PYs = Person-years; Rate: per 100,000 PYs; Adjusted HR (hazard ratio): Adjusted variables listed in the table; CI = confidence interval

*P < 0.05, **P < 0.01

***P < 0.001

**Table 5. Factors of sensorineural hearing loss by using Cox regression and Fine & Gray's competing risk model.**

| | No competing risk in the model | | | | | Competing risk in the model | | | | |
|---|---|---|---|---|---|---|---|---|---|---|
| | Wald | Adjusted HR | 95% CI | 95% CI | P | Wald | Adjusted HR | 95% CI | 95% CI | P |
| HTN × CKD | 18.29 | 2.37 | 2.00 | 2.81 | <0.001*** | 18.16 | 2.37 | 2.00 | 2.80 | <0.001*** |
| T2DM × CKD | 4.01 | 1.32 | 1.01 | 1.73 | 0.045* | 3.80 | 1.31 | 1.02 | 1.71 | 0.048* |
| HF × CKD | 6.66 | 1.78 | 1.24 | 2.56 | 0.012* | 9.39 | 1.92 | 1.34 | 2.76 | 0.005** |
| Stroke × CKD | 32.18 | 2.52 | 1.83 | 3.47 | <0.001*** | 38.00 | 2.73 | 1.99 | 3.76 | <0.001*** |
| COPD × CKD | 19.27 | 2.39 | 1.62 | 3.53 | <0.001*** | 22.57 | 2.60 | 1.74 | 3.79 | <0.001*** |
| Liver cirrhosis × CKD | 23.16 | 2.70 | 1.80 | 4.05 | <0.001*** | 27.86 | 2.98 | 1.99 | 4.46 | <0.001*** |
| Meniere's disease × CKD | 54.84 | 8.59 | 4.86 | 15.15 | <0.001*** | 49.34 | 7.69 | 4.35 | 13.58 | <0.001*** |
| Aminoglycoside × CKD | 0.84 | 1.24 | 0.68 | 1.86 | 0.416 | 0.96 | 1.46 | 0.75 | 1.99 | 0.375 |
| Loop diuretics × CKD | 0.75 | 1.37 | 0.72 | 2.04 | 0.529 | 0.83 | 1.59 | 0.89 | 2.11 | 0.428 |

Adjusted HR (hazard ratio): Adjusted variables listed in Table 2; CI = confidence interval

*$P < 0.05$

**$P < 0.01$

***$P < 0.001$

Reference: Without

disequilibrium of endolymph, ischemia and subsequent reperfusion may lead to the hearing deficiency associated with dialysis [26, 27]. Because of severe CKD and influence of hemodialysis, the higher incidence of SNHL in CKD patients with hemodialysis was noticed. Notably, in this study, having CKD for longer time was associated with more significantly heightened incidence of developing SNHL (Fig 2). The similar findings were also noticed in patients with type 2 DM, cardiovascular diseases and COPD [28–36]. Time as a risk factor was also confirmed in CKD animal models, where the impairment of cochlear function was exacerbated over time [19, 37]. Furthermore, SNHL is also associated with tinnitus [21], as diminished output from the damaged cochlea causes an increased spontaneous activity in the dorsal cochlear nucleus [38, 39].

CKD patients often have multiple systemic dysfunctions such as cardiovascular, lung, liver, metabolic, brain, immune system, and chronic inflammation further resulting in various comorbidities [40–42]. Cardiovascular disease, type 2 DM and liver cirrhosis have been strongly considered as risk factors of CKD because of inflammation, ischemia, hemodynamic change, and the overactivity of renin–angiotensin–aldosterone and sympathetic nervous systems [43–50]. In recent studies, COPD may be another risk factor of CKD while it is a disease with irreversible airway obstruction and contributes to systemic inflammation that may induce vessel disease [51, 52]. Furthermore, there was significant risk of type 2 DM for SNHL [28–29]. There were higher values of HR in our patients with cardiovascular diseases such as stroke. Coronary artery disease was linked to the occurrence of SNHL [19, 30]. Cardiovascular disease may be an important factor of developing SNHL [30–34]. Recent reports presented the close relation between SNHL and COPD [35]. Additionally, smoking was a risk factor of SNHL in patients with COPD [36]. There was no study indicating the association between liver cirrhosis and SNHL. However, our result presented no significant correlation between them. In contrast, previous studies showed various effects of different causes of liver cirrhosis for SNHL such as hepatitis B and C increased the risk [53, 54], while a person with drinking alcohol habit had less chances [55]. (Table 2).

Our study highlighted the significant interaction of developing SNHL in patients with CKD and comorbidities such as stroke, liver cirrhosis, COPD, hypertension, HF, and type 2 DM

(Table 5). Organ crosstalk is essential for maintaining physical homeostasis; however, dysfunction in one or more organs may lead to functional and structural pathological states in other organs. CKD may be a factor to aggravate the synergetic effect of developing SNHL in patients with other systemic diseases by inter-organ crosstalk, which may contribute to chronic inflammation, oxidative stress, and sympathetic nerve hyperactivity [43, 48, 50, 51, 54, 56]. The future research for underlying mechanism is necessary to preserve hearing function.

The strengths of this study are its population-based research, use of well-established cohort data with a large sample size, and extended follow-up period to identify CKD as a risk factor for developing SNHL. However, this study also has few limitations. Firstly, the lack of comparison for several possible other risk factors such as smoking history and noise exposure may create bias. Secondly, although NHIRD coding in recording diseases has been validated [12, 13], there is no data regarding laboratory and audiometry parameters for coding the accuracy and accessing the severity of diseases. Nevertheless, in an attempt to increase the validity of diagnosis, this study matched diagnosis to three indices and restricted the diagnosis of hearing loss by an otorhinolaryngologist only. The severity of CKD with and without hemodialysis for SNHL was identified (Table 3). However, the influence of the stage 1–5 of CKD on the degree and type (low or high frequency) of SNHL cannot be analyzed. Thirdly, there was an association between different routes of administration of loop diuretic and ototoxicity [57]. However, we were unable to compare the influence of oral and intravenous loop diuretic therapy since sequential intravenous-to-oral and oral-to-intravenous switch regimens were often used in clinical practice. Moreover, the other ototoxic drugs or genetic effects [58] that were not included may contribute bias. Finally, we did not investigate the underlying pathophysiological mechanism associating CKD with SNHL. Future work and analysis are needed to understand the occurrence of SNHL in this high-risk population.

## Conclusion

This study revealed that the incidence of developing SNHL was higher in patients with CKD and that this incidence considerably increased with CKD duration. Having CKD with comorbidities increased the interaction of developing hearing loss. Based on our results, the future study of mechanism in this high-risk population is necessary to develop effective strategies of hearing protection. Collectively, these findings provide important and required insight into the relationship between SNHL and associated comorbidities in CKD patients.

## Author Contributions

**Conceptualization:** Kun-Lin Wu, Cheng-Ping Shih, Wu-Chien Chien, Po-Jen Hsiao.

**Data curation:** Jenq-Shyong Chan, Chi-Hsiang Chung, Hung-Che Lin, Chang-Huei Tsao, Fu-Huang Lin, Wu-Chien Chien.

**Formal analysis:** Kun-Lin Wu, Chi-Hsiang Chung, Hung-Che Lin, Chang-Huei Tsao, Fu-Huang Lin, Wu-Chien Chien, Po-Jen Hsiao.

**Funding acquisition:** Kun-Lin Wu, Jenq-Shyong Chan.

**Investigation:** Kun-Lin Wu, Cheng-Ping Shih, Jenq-Shyong Chan, Chi-Hsiang Chung, Wu-Chien Chien, Po-Jen Hsiao.

**Methodology:** Cheng-Ping Shih, Chi-Hsiang Chung, Hung-Che Lin, Chang-Huei Tsao, Fu-Huang Lin, Wu-Chien Chien, Po-Jen Hsiao.

**Project administration:** Wu-Chien Chien.

**Resources:** Po-Jen Hsiao.

**Software:** Chi-Hsiang Chung.

**Supervision:** Jenq-Shyong Chan, Wu-Chien Chien, Po-Jen Hsiao.

**Validation:** Po-Jen Hsiao.

**Writing – original draft:** Kun-Lin Wu.

**Writing – review & editing:** Wu-Chien Chien, Po-Jen Hsiao.

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
