## [Decision Letter · Decision Letter 0]

12 May 2020

PONE-D-19-29457

Investigation of the relationship between sensorineural hearing loss and associated comorbidities in patients with chronic kidney disease: a nationwide, population-based cohort study

PLOS ONE

Dear Dr. Hsiao,

Thank you for submitting your manuscript to PLOS ONE. After careful consideration, we feel that it has merit but does not fully meet PLOS ONE’s publication criteria as it currently stands. Therefore, we invite you to submit a revised version of the manuscript that addresses the points raised during the review process.

We would appreciate receiving your revised manuscript by Jun 25 2020 11:59PM. To enhance the reproducibility of your results, we recommend that if applicable you deposit your laboratory protocols in protocols.io, where a protocol can be assigned its own identifier (DOI) such that it can be cited independently in the future. For instructions see: http://journals.plos.org/plosone/s/submission-guidelines#loc-laboratory-protocols

We look forward to receiving your revised manuscript.

Kind regards,

Natasha McDonald

Associate Editor

PLOS ONE

Journal Requirements:

Reviewers' comments:

Reviewer's Responses to Questions

**Comments to the Author**

1. Is the manuscript technically sound, and do the data support the conclusions?

Reviewer #1: Partly

Reviewer #2: No

Reviewer #3: Yes

2. Has the statistical analysis been performed appropriately and rigorously? 

Reviewer #1: No

Reviewer #2: Yes

Reviewer #3: Yes

3. Have the authors made all data underlying the findings in their manuscript fully available?

Reviewer #1: Yes

Reviewer #2: No

Reviewer #3: No

4. Is the manuscript presented in an intelligible fashion and written in standard English?

Reviewer #1: Yes

Reviewer #2: Yes

Reviewer #3: Yes

5. Review Comments to the Author

Reviewer #1: In this paper the authors investigate the relationship between sensorineural hearing loss and associated

comorbidities in patients with chronic kidney disease.

The study is retrospective and population-based, and the number of enrolled subjects is the major strength.

However, some issues need to be clarified:

1. It is not clear why only few cardiovascular comorbidities were considered in the models. Authors recognize the importance of ischaemic events, but coronary heart disease, cardiac ischemia or myocardial infarction are not included. A possible explanation is that they are included in the CCI, but even in this case, including in a model such score together with the other comorbidities there is the risk of co-linearity. A possible explanation could be that all these variables have been removed to calculate CCI_R, but in this case it is not clear what this index contains. Please explain.

2. The author attest that CKD is the variable which influences the most the onset of sensorineural hearing loss, because the HR of CKD is higher than HRs of the other variables included in the model. However this is not totally true, as in order to score the importance of one variable the Wald coefficient must be taken into account. Please report this value in the tables.

3. Figure 2 reports Kaplan-Meier for cumulative risk of sensorineural hearing loss among patients aged 18 and over stratified by CKD with log-rank test in competing risk model. The end the curves are quite peculiar, as they look like a huge number of events occurred during last year of follow-up. Has proportionality of risk been assessed? Please explain. Secondly, it is possible to use a sort of K-M analysis to deal with competing risk (the name is cumulative incidence competing risk (CICR) method), but anyway log-rank test cannot be used to compare the curves (Verduijn M, Grootendorst DC, Dekker FW et al. The analysis of competing events like cause-specific mortality—beware of the Kaplan–Meier method. Nephrol Dial Transplant 2011; 26: 56–61; Bland JM, Altman DG. The logrank test. BMJ 2004; 328: 1073).

4. In table 3 it is not clear how models were built. It is confusing to have in the same table no CKD, CKD and CKD with/without haemodialysis. What has been compared to what?

5. At page 14, line 197, authors write “The increased values of adjusted HRs indicated synergetic effect of CKD and comorbidities for SNHL”. This is not the correct approach to understand if two variables act sinergically on a specific outcome, as the increase must be higher than the sum of HRs of the single variables (additive interaction) or that their product (multiplicative interaction). The easiest way to assess if a variable act on another one to determine a specific outcome is to introduce in the model a multiplicative term (i.e., if I want to assess the interaction between CKD and DM, I enter in the model CKD, DM and CKD*DM). I suggest to perform again the analysis by using this approach rather than a simple stratification.

Minor comments:

I suggest to remove the word “matched” from the description of the cohort, as it evokes case-control studies and another type of analysis.

In Table 2, please unify the columns of 95% CI

It is not CKD which compete with the risk of death, but hearing loss.

Reviewer #2: General: The main reservation I have is that association does not necessarily imply causality, and while these associations may be regarded as statistically relevant and likely important, the authors have not demonstrated causality or proved that they are “risk factors”.

Lines 88-101 authors evaluate comorbidities to “identify the risk of developing SHML”. What the authors evaluated was statistical association; they have not evaluated or demonstrated causality, and therefore “risk factors”. This interpretation is repeated multiple times throughout the manuscript (line 114). In assessing the importance of comorbidities in SNHL it is important to identify statistical associations (which the authors did in this study); the next step is to identify possible causality (mechanisms of disease that can be acted upon); followed by intervention and prevention/therapy. This study can only claim the first step. There are well documented mechanisms why CKD can result in SNHL; mechanisms have not been evaluated for comorbidities and SNHL. Comorbidities were more common in the group with CKD; undoubtedly, SNHL will be statistically more common in patients who take the elevator to the CJD clinic; however, although it is a statistically significant association, the elevator to the clinic represents no causality on SNHL. It is true that comorbidities evaluated are a risk factor for CKD through many different mechanisms. In order to understand the role of comorbidities in the pathogenesis of SNHL it would be important to evaluate the presence of SNHL in patients with these comorbidities WITHOUT chronic kidney disease. The authors’ large database should provide this information. “Inter-organ cross talk” is a very vague definition that does not provide a well-defined pathogenetic mechanism.

Was there any correlation between the duration of comorbidities or their severity and SNHL?

Sentences in lines 240-241 should be rephrased and misspelling (controversal) corrected.

Sentences in lines 268-269 should be rephrased.

Lines 204-205 authors state It was observed that the incidence of CKD was 25%. What does this mean?

The authors evaluated the use of aminoglycosides and loop diuretics; what about use of other ototoxic drugs?

Why the authors evaluate monthly income, and was there any association?

Can the authors evaluate if there is an association between the severity of the chronic kidney disease and the presence of SNHL?

Specific: Lines 88 and 90: both the organs – should read: both organs.

Line 95 …have been showed… should read …have been shown…

Reviewer #3: Dear Authors,

Congratulations on a well written manuscript.

This is a well written paper on an increasingly recognised association between CKD and hearing loss. The sample size is quite large and despite retrospective nature, authors have used rigorous statistical methods including multivariate regression to adjust for confounders.

However, from an ENT perspective, I would be curious to know how the SNHL was classified in the patients (degree of hearing loss) and whether the association varied with the severity as well as the type (high vs low frequency) of SNHL. These factors might be important to the clinician.

6. PLOS authors have the option to publish the peer review history of their article (what does this mean?). If published, this will include your full peer review and any attached files.

Reviewer #1: No

Reviewer #2: No

Reviewer #3: Yes: Jishana Jamaldeen

---

## [Author Response · Author response to Decision Letter 0]

29 Jun 2020

Response to Reviewer 1:

General comments 

In this paper the authors investigate the relationship between sensorineural hearing loss and associated comorbidities in patients with chronic kidney disease. The study is retrospective and population-based, and the number of enrolled subjects is the major strength.

Remedy: We are grateful for your positive comments and careful reading of our manuscript. We have revised our manuscript according to your comments and the corresponding changes in the revised manuscript have been highlighted in red font. The itemized responses to your comments are below.

Major Comments: 

Point 1. It is not clear why only few cardiovascular comorbidities were considered in the models. Authors recognize the importance of ischaemic events, but coronary heart disease, cardiac ischemia or myocardial infarction are not included. A possible explanation is that they are included in the CCI, but even in this case, including in a model such score together with the other comorbidities there is the risk of co-linearity. A possible explanation could be that all these variables have been removed to calculate CCI_R, but in this case it is not clear what this index contains. Please explain.

Remedy: We thank the valuable comment. The cardiovascular diseases are a group of disorders of the heart and blood vessels. We had analyzed the correlation between SNHL and cardiovascular diseases such as heart failure (HF), stroke and HTN in our study. The link between severity of coronary artery disease and degree of sensorineural hearing loss had been reported [30]. We revised the sentence, “There were higher values of HR in our patients with cardiovascular diseases such as stroke. Coronary artery disease was linked to the occurrence of SNHL [19,30]. Cardiovascular disease may be an important factor of developing SNHL [30-34].” (please see Discussion, Lines 243-246)

In Charlson Comorbidity Index (CCI), there are 19 indicators of 17 disease including myocardial infarction, congestive heart failure, peripheral vascular disease, CVA or TIA, dementia, COPD, connective tissue disease, peptic ulcer disease, liver disease, diabetes mellitus, hemiplegia, moderate to severe chronic kidney disease, solid tumor with or without metastasis, leukemia, lymphoma and AIDS. We used CCI_R after removing CKD, type 2 DM, CHF, stroke, COPD, and liver cirrhosis to evaluate the risk of developing SNHL. In our manuscript, “The 19 weighted indicators of 17 comorbidities were used to calculate the Charlson Comorbidity Index (CCI) [16]. The variables, including CKD, type 2 DM, HF, stroke, COPD, and liver cirrhosis, have been removed to calculate CCI_R.” (please see Materials and Methods, Lines 157-159)

Reference 

30. Erkan AF, Beriat GK, Ekici B, Doğan C, Kocatürk S, Töre HF. Link between angiographic extent and severity of coronary artery disease and degree of sensorineural hearing loss. Herz. 2015;40:481-6.

Point 2. The author attest that CKD is the variable which influences the most the onset of sensorineural hearing loss, because the HR of CKD is higher than HRs of the other variables included in the model. However, this is not totally true, as in order to score the importance of one variable the Wald coefficient must be taken into account. Please report this value in the tables.

Remedy: We completely agree with this comment. We have added Wald coefficient in Tables 2-5 to score the influence of variables. 

Point 3. Figure 2 reports Kaplan-Meier for cumulative risk of sensorineural hearing loss among patients aged 18 and over stratified by CKD with log-rank test in competing risk model. The end the curves are quite peculiar, as they look like a huge number of events occurred during last year of follow-up. Has proportionality of risk been assessed? Please explain. Secondly, it is possible to use a sort of K-M analysis to deal with competing risk (the name is cumulative incidence competing risk (CICR) method), but anyway log-rank test cannot be used to compare the curves (Verduijn M, Grootendorst DC, Dekker FW et al. The analysis of competing events like cause-specific mortality—beware of the Kaplan–Meier method. Nephrol Dial Transplant 2011; 26: 56–61; Bland JM, Altman DG. The logrank test. BMJ 2004; 328: 1073).

Remedy: We completely agree with this invaluable comment. Our study is a 10-year retrospective cohort study. We have made the corrections immediately and evaluated the risk of sensorineural hearing loss among patients aged 18 and over stratified by CKD with cumulative incidence competing risk (CICR) method and presented the results proportionally (Figure 2). The sentences were revised, “The cumulative incidence competing risk (CICR) method was used to estimate the difference in the risk of developing hearing loss between the CKD and comparison groups [15].” (Materials and Methods, line 154-156) and “The cumulative incidence competing risk (CICR) analysis indicated that patients with CKD had a significantly higher incidence of developing SNHL over time than comparison participants (p <.001) (Fig 2). “(please see Results, Lines 178-181)

Reference: 

15. Verduijn M, Grootendorst DC, Dekker FW, Jager KJ, le Cessie S. The analysis of competing events like cause-specific mortality--beware of the Kaplan-Meier method. Nephrol Dial Transplant. 2011;26:56-61.

Figure 2. The cumulative incidence competing risk (CICR) method for the incidence of sensorineural hearing loss among patients aged 18 and over stratified by CKD (p <.001).

Point 4. In table 3 it is not clear how models were built. It is confusing to have in the same table no CKD, CKD and CKD with/without haemodialysis. What has been compared to what?

Remedy: We greatly appreciate this comment. The patients with CKD classified by ICD-9 included ones with ESRD who received hemodialysis and ones with CKD who did not receive hemodialysis. In previous study, hemodialysis is a risk factor for developing SNHL. Osmotic disequilibrium of endolymph, ischemia and subsequent reperfusion may lead to the hearing deficiency associated with dialysis. We aimed to evaluate this association. The advanced analysis was performed to differentiate it and presented in Table 3.

Point 5. At page 14, line 197, authors write “The increased values of adjusted HRs indicated synergetic effect of CKD and comorbidities for SNHL”. This is not the correct approach to understand if two variables act sinergically on a specific outcome, as the increase must be higher than the sum of HRs of the single variables (additive interaction) or that their product (multiplicative interaction). The easiest way to assess if a variable act on another one to determine a specific outcome is to introduce in the model a multiplicative term (i.e., if I want to assess the interaction between CKD and DM, I enter in the model CKD, DM and CKD*DM). I suggest to perform again the analysis by using this approach rather than a simple stratification.

Remedy: The authors thank to the valuable suggestion and amend our manuscript. The interaction of CKD and comorbidities for SNHL has been analyzed in Table 5. We amend our manuscript “We next focus our investigation on identifying and quantifying the multiplicative interaction of comorbidities on CKD. In Table 5, the highest Wald coefficient and adjusted HR were 49.34 and 7.69 in those with Meniere’s disease, 38 and 2.73 in those with stroke, 27.86 and 2.98 in those with liver cirrhosis, 22.57 and 2.6 in those with COPD, 18.16 and 2.37 in those with HTN, 9.39 and 1.92 in those with CHF, and 3.8 and 1.31 in those with Type 2 DM, respectively (Table 5). Interestingly, we demonstrated that the interactions of comorbidities on CKD was significant for SNHL.” (please see Results: Lines 198-202)

Minor comments

Point 1. I suggest to remove the word “matched” from the description of the cohort, as it evokes case-control studies and another type of analysis.

Remedy:

The authors thank to the valuable suggestion and we have removed “matched”. (please see Lines 102, 120, 146)

Point 2. In Table 2, please unify the columns of 95% CI

Remedy:

The authors thank to the valuable suggestion and unified the columns of 95% CI.

Point 3. It is not CKD which compete with the risk of death, but hearing loss.

Remedy: We greatly appreciate this comment. The sentence has been corrected that “We also performed a competing-risks regression (Fine-Gray model) because SNHL risk might compete with the risk of death” (please see Lines 152-154)

Last, we are deeply honored by the time and effort you spent in reviewing this manuscript. In reviewing and revising our manuscript, we are motivated to read more and thus learn more from your criticisms.

Response to Reviewer 2:

General comments: 

The main reservation I have is that association does not necessarily imply causality, and while these associations may be regarded as statistically relevant and likely important, the authors have not demonstrated causality or proved that they are “risk factors”.

Remedy: We are grateful for your positive comments and careful reading of our manuscript. We have revised our manuscript according to your comments and the corresponding changes in the revised manuscript have been highlighted in red font. The itemized responses to your comments are below.

Major concerns: 

Point 1. Lines 88-101 authors evaluate comorbidities to “identify the risk of developing SHML”. What the authors evaluated was statistical association; they have not evaluated or demonstrated causality, and therefore “risk factors”. This interpretation is repeated multiple times throughout the manuscript (line 114). In assessing the importance of comorbidities in SNHL it is important to identify statistical associations (which the authors did in this study); the next step is to identify possible causality (mechanisms of disease that can be acted upon); followed by intervention and prevention/therapy. This study can only claim the first step. There are well documented mechanisms why CKD can result in SNHL; mechanisms have not been evaluated for comorbidities and SNHL. Comorbidities were more common in the group with CKD; undoubtedly, SNHL will be statistically more common in patients who take the elevator to the CKD clinic; however, although it is a statistically significant association, the elevator to the clinic represents no causality on SNHL. It is true that comorbidities evaluated are a risk factor for CKD through many different mechanisms. In order to understand the role of comorbidities in the pathogenesis of SNHL it would be important to evaluate the presence of SNHL in patients with these comorbidities WITHOUT chronic kidney disease. The authors’ large database should provide this information. “Inter-organ cross talk” is a very vague definition that does not provide a well-defined pathogenetic mechanism.

Remedy: We fully agreed your concern. The correlations between CKD and associated comorbidities had been reported in previous studies. The relation of hearing loss and CKD and other systemic diseases had also been demonstrated. We have designed the advanced study to investigate the relationship of hearing loss in patients with CKD and associated comorbidities. Taiwan NHIRD was used as the database. We avoided to use “risk” in our results. The multiplicative interaction of comorbidities on CKD for hearing loss revealed statistical significance of increased adjusted HRs in Table 5. “Inter-organ cross talk” between kidney and other organs may be a mechanism for this issue, but more research and analysis are needed to understand the occurrence of SNHL in this high-risk population. We have revised the paragraph, “Our study highlighted the significant interaction of developing SNHL in patients with CKD and comorbidities such as stroke, liver cirrhosis, COPD, hypertension, HF, and type 2 DM (table 5). Organ crosstalk is essential for maintaining physical homeostasis; however, dysfunction in one or more organs may lead to functional and structural pathological states in other organs. CKD may be a factor to aggravate the synergetic effect of developing SNHL in patients with other systemic diseases by inter-organ crosstalk, which may contribute to chronic inflammation, oxidative stress, and sympathetic nerve hyperactivity [43,48,50,51,54,56]. The future research for underlying mechanism is necessary to preserve hearing function. “ (please see Discussion: Lines: 254-262)

Point 2. Was there any correlation between the duration of comorbidities or their severity and SNHL?

Remedy: Thank you for your suggestion. We focused on the influence of CKD with comorbidities. The correlation between duration of CKD and hearing loss was analyzed in our studies. In previous studies, the significant result of the duration of comorbidities in SNHL had been demonstrated [1-8]. In National Health Insurance database (NHIRD) of Taiwan, there is no data regarding laboratory and audiometry parameters for accessing the severity of diseases. The influence of the severity of diseases on the severity and type (low or high frequency) of SNHL cannot be analyzed. Nevertheless, in an attempt to evaluate the severity of CKD for SNHL, CKD with and without hemodialysis for SNHL were analyzed in Table 3. We amend the sentences “Because of severe CKD and influence of hemodialysis, the higher incidence of SNHL in CKD patients with hemodialysis was noticed.” (please see Discussion: Lines: 225-227), “Notably, in this study, having CKD for longer time was associated with more significantly heightened incidence of developing SNHL (Figure 2). The similar findings were also noticed in patients with type 2 DM, cardiovascular diseases and COPD [28-36].” (please see Discussion: Lines: 227-230), and “Secondly, although NHIRD coding in recording diseases has been validated [12,13], there is no data regarding laboratory and audiometry parameters for coding the accuracy and accessing the severity of diseases. Nevertheless, in an attempt to increase the validity of diagnosis, this study matched diagnosis to three indices and restricted the diagnosis of hearing loss by an otorhinolaryngologist only. The severity of CKD with and without hemodialysis for SNHL was identified (Table 3). However, the influence of the stage 1-5 of CKD on the degree and type (low or high frequency) of SNHL cannot be analyzed.” (please see Discussion: Lines: 267-275)

1. Sterling MR, Lin FR, Jannat-Khah DP, Goman AM, Echeverria SE, Safford MM. Hearing Loss Among Older Adults With Heart Failure in the United States: Data From the National Health and Nutrition Examination Survey. JAMA Otolaryngol Head Neck Surg. 2018;144:273-5. 

2. Schade DS, Lorenzi GM, Braffett BH, Gao X, Bainbridge KE, Barnie A, et al. Hearing Impairment and Type 1 Diabetes in the Diabetes Control and Complications Trial/Epidemiology of Diabetes Interventions and Complications (DCCT/EDIC) Cohort. Diabetes Care. 2018;41:2495-501.

3. Helzner EP, Contrera KJ. Type 2 Diabetes and Hearing Impairment. Curr Diab Rep. 2016;16:3. 

4. Sterling MR, Silva AF, Charlson ME. Sensory Impairments in Heart Failure-Are We Missing the Basics?: A Teachable Moment. JAMA Intern Med. 2018;178:843-44. 

5. Agrawal Y, Platz EA, Niparko JK. Prevalence of hearing loss and differences by demographic characteristics among US adults: data from the National Health and Nutrition Examination Survey, 1999-2004. Arch Intern Med. 2008;168:1522-30. 

6. Gates GA, Cobb JL, D'Agostino RB, Wolf PA. The relation of hearing in the elderly to the presence of cardiovascular disease and cardiovascular risk factors. Arch Otolaryngol Head Neck Surg. 1993;119:156-61.

7. Arash Bayat, Nader Saki, Soheila Nikakhlagh, Golshan Mirmomeni, Hanieh Raji, et al. Is COPD associated with alterations in hearing? A systematic review and meta-analysis. Int J Chron Obstruct Pulmon Dis. 2019;14:149-62. 

8. Kamenski G, Bendova J, Fink W, Sönnichsen A, Spiegel W, Zehetmayer S. Does COPD have a clinically relevant impact on hearing loss? A retrospective matched cohort study with selection of patients diagnosed with COPD. BMJ Open. 2015;5:e008247. 

Point 3. Sentences in lines 240-241 should be rephrased and misspelling (controversal) corrected.

Remedy: We revised these sentences, “However, our result presented no significant correlation between them. In contrast, previous studies showed various effects of different causes of liver cirrhosis for SNHL such as hepatitis B and C increased the risk [53,54], while a person with drinking alcohol habit had less chances [55]. (Table 2).” (please see Discussion: Lines 249-253)

Point 4. Sentences in lines 268-269 should be rephrased.

Remedy: Thank your suggestion. We revised this sentence,” Future work and analysis are needed to understand the occurrence of SNHL in this high-risk population. 

” (please see Discussion: Lines 281-282)

Point 5. Lines 204-205 authors state It was observed that the incidence of CKD was 25%. What does this mean?

Remedy: Thank you for the reminder. In our study, the prevalence of CKD was 25% of patients receiving out-patient care which was higher than 13-15% of general population. We corrected it “It was observed that the prevalence of CKD was 25% of patients receiving out-patient care, particularly in male ones.” (please see Discussion: Lines 207-208)

Point 6. The authors evaluated the use of aminoglycosides and loop diuretics; what about use of other ototoxic drugs?

Remedy: Thank your command. In patients with CKD, aminoglycoside and loop diuretics are commonly used to control infection and fluid status, so we tried to evaluate the effect. However, other ototoxic drugs were not included in our study. So, we added the sentence that “Moreover, the other ototoxic drugs or genetic effects that were not included may contribute bias.” in the limitation of discussion. 

(please see Discussion: Lines 277-278)

Point 7. Why the authors evaluate monthly income, and was there any association?

Remedy: Thank your suggestion. The insured premium of National Health Insurance Program was set by our government according to monthly income. Higher incurred premium means higher social-economic level in Taiwan. The acceptable burden of insured premium was designed to cover medical expenses of more than 99% of the 23 million inhabitants. Our study reviewed the similar socio-economic distribution in patients with CKD and without CKD. We revised it, “The covariates of gender, age groups (18–29, 30–39, 40–49, 50–59, ≥60 years), and insured premium [in New Taiwan Dollars; <18,000, 18,000–34,999, ≥35,000] were analyzed.” (please see Materials and Methods: Lines 137-139)

Point 8. Can the authors evaluate if there is an association between the severity of the chronic kidney disease and the presence of SNHL?

Remedy: Thank your suggestion. The ICD-9 code of disease was recorded in NHIRD. The stage 1-5 of CKD and severity and type of hearing loss could not be evaluated in our study. However, the severity of CKD with and without hemodialysis was identified to evaluate the influence of SNHL in Table 3. We descripted this issue in discussion, “Because of severe CKD and influence of hemodialysis, the higher incidence of SNHL in CKD patients with hemodialysis was noticed.” (please see Discussion: Lines: 225-227), and “Secondly, although NHIRD coding in recording diseases has been validated [12,13], there is no data regarding laboratory and audiometry parameters for coding the accuracy and accessing the severity of diseases. Nevertheless, in an attempt to increase the validity of diagnosis, this study matched diagnosis to three indices and restricted the diagnosis of hearing loss by an otorhinolaryngologist only. The severity of CKD with and without hemodialysis for SNHL was identified (Table 3). However, the influence of the stage 1-5 of CKD on the degree and type (low or high frequency) of SNHL cannot be analyzed.” (please see Discussion: Lines: 267-275)

Point 9. Specific: Lines 88 and 90: both the organs – should read: both organs.

Remedy: Thank your suggestion. We made the corrections immediately. (please see Discussion: Lines 88 and 90)

Point 10. Line 95 …have been showed… should read …have been shown…

Remedy: Thank your suggestion. We made the corrections immediately. (please see Discussion: Line 95)

Last, I am deeply honored by the time and effort you spent in reviewing this manuscript. In reviewing and revising our manuscript, I was motivated to read more and thus learn more from your criticisms.

Response to Reviewer 3:

General comments 

Dear Authors,

Congratulations on a well written manuscript.

This is a well written paper on an increasingly recognised association between CKD and hearing loss. The sample size is quite large and despite retrospective nature, authors have used rigorous statistical methods including multivariate regression to adjust for confounders.

Remedy: We are grateful for your positive comments and careful reading of our manuscript. We have revised our manuscript according to your comments and the corresponding changes in the revised manuscript have been highlighted in red font. The itemized responses to your comments are below.

Major Comments: 

Point 1. However, from an ENT perspective, I would be curious to know how the SNHL was classified in the patients (degree of hearing loss) and whether the association varied with the severity as well as the type (high vs low frequency) of SNHL. These factors might be important to the clinician.

Remedy: We thank the valuable comment. In NHIRD, the ICD-9 code of disease was recorded without parameters of laboratory examination and audiometry. The association of stage 1-5 of CKD and severity and type of hearing loss could not be evaluated in our study. However, the severity of CKD with and without hemodialysis was identified to evaluate the influence of SNHL in Table 3. We descripted this issue in discussion, “Because of severe CKD and influence of hemodialysis, the higher incidence of SNHL in CKD patients with hemodialysis was noticed.” (please see Discussion: Lines: 225-227), and “Secondly, although NHIRD coding in recording diseases has been validated [12,13], there is no data regarding laboratory and audiometry parameters for coding the accuracy and accessing the severity of diseases. Nevertheless, in an attempt to increase the validity of diagnosis, this study matched diagnosis to three indices and restricted the diagnosis of hearing loss by an otorhinolaryngologist only. The severity of CKD with and without hemodialysis for SNHL was identified (Table 3). However, the influence of the stage 1-5 of CKD on the degree and type (low or high frequency) of SNHL cannot be analyzed.” (please see Discussion: Lines: 267-275)

Last, I am deeply honored by the time and effort you spent in reviewing this manuscript. In reviewing and revising our manuscript, I was motivated to read more and thus learn more from your criticisms.

---

## [Decision Letter · Decision Letter 1]

27 Aug 2020

Investigation of the relationship between sensorineural hearing loss and associated comorbidities in patients with chronic kidney disease: a nationwide, population-based cohort study

PONE-D-19-29457R1

Dear Dr. Hsiao,

We’re pleased to inform you that your manuscript has been judged scientifically suitable for publication and will be formally accepted for publication once it meets all outstanding technical requirements.

Kind regards,

CLAUDIA TORINO

Guest Editor

PLOS ONE

Additional Editor Comments (optional):

Reviewers' comments:

Reviewer's Responses to Questions

**Comments to the Author**

1. If the authors have adequately addressed your comments raised in a previous round of review and you feel that this manuscript is now acceptable for publication, you may indicate that here to bypass the “Comments to the Author” section, enter your conflict of interest statement in the “Confidential to Editor” section, and submit your "Accept" recommendation.

Reviewer #1: All comments have been addressed

Reviewer #4: All comments have been addressed

2. Is the manuscript technically sound, and do the data support the conclusions?

Reviewer #1: Yes

Reviewer #4: Yes

3. Has the statistical analysis been performed appropriately and rigorously? 

Reviewer #1: Yes

Reviewer #4: Yes

4. Have the authors made all data underlying the findings in their manuscript fully available?

Reviewer #1: No

Reviewer #4: Yes

5. Is the manuscript presented in an intelligible fashion and written in standard English?

Reviewer #1: Yes

Reviewer #4: Yes

6. Review Comments to the Author

Reviewer #1: Dear authors, I thank you to have carefully addressed all the raised issues. I have no further comments.

Reviewer #4: Dear authors,

Thank you for your efforts. All points were tackled in a professional way. You answered all comments by reviewers and the current version of the manuscript is better to e considered for publication

7. PLOS authors have the option to publish the peer review history of their article (what does this mean?). If published, this will include your full peer review and any attached files.

Reviewer #1: No

Reviewer #4: No

---

## [Editor Report · Acceptance letter]

3 Sep 2020

PONE-D-19-29457R1 

Investigation of the relationship between sensorineural hearing loss and associated comorbidities in patients with chronic kidney disease: a nationwide, population-based cohort study 

Dear Dr. Hsiao:

I'm pleased to inform you that your manuscript has been deemed suitable for publication in PLOS ONE. Congratulations! Your manuscript is now with our production department. 

Kind regards, 

on behalf of

Dr. CLAUDIA TORINO 

Guest Editor

PLOS ONE